# Solving the Right Problem is Key for Translational NLP:
# A Case Study in UMLS Vocabulary Insertion

**Bernal Jiménez Gutiérrez**[1,*]**, Yuqing Mao**[2]**, Vinh Nguyen**[2]**,**
**Kin Wah Fung**[2]**, Yu Su**[1]**, Olivier Bodenreider**[2]

[1]The Ohio State University, [2]National Library of Medicine

{jimenezgutierrez.1,su.809}@osu.edu

{yuqing.mao,vinh.nguyen,kwfung}@nih.gov, olivier@nlm.nih.gov

## Abstract

As the immense opportunities enabled by large language models become more apparent, NLP systems will be increasingly expected to excel in real-world settings. However, in many instances, powerful models alone will not yield translational NLP solutions, especially if the formulated problem is not well aligned with the real-world task. In this work, we study the case of UMLS vocabulary insertion, an important real-world task in which hundreds of thousands of new terms, referred to as atoms, are added to the UMLS, one of the most comprehensive open-source biomedical knowledge bases (Bodenreider, 2004). Previous work aimed to develop an automated NLP system to make this time-consuming, costly, and error-prone task more efficient. Nevertheless, practical progress in this direction has been difficult to achieve due to a problem formulation and evaluation gap between research output and the real-world task. In order to address this gap, we introduce a new formulation for UMLS vocabulary insertion which mirrors the real-world task, datasets which faithfully represent it and several strong baselines we developed through re-purposing existing solutions. Additionally, we propose an effective rule-enhanced biomedical language model which enables important new model behavior, outperforms all strong baselines and provides measurable qualitative improvements to editors who carry out the UVI task. We hope this case study provides insight into the considerable importance of problem formulation for the success of translational NLP solutions.[1]

## 1 Introduction

The public release of large language model (LLM) products like ChatGPT has triggered a wave of enthusiasm for NLP technologies. As more people discover the wealth of opportunities enabled by

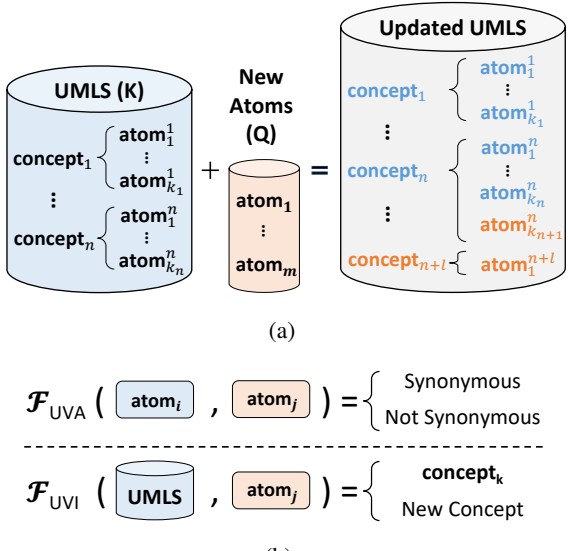

Figure 1: The UMLS update process in (a) introduces atoms from individual sources into the original UMLS as synonyms of existing concepts or entirely new concepts. The UVA task is formulated as binary synonymy prediction (b) and was thus unable to tackle the real-world update task addressed by our UVI formulation.

these technologies, NLP systems will be expected to perform in a wide variety of real-world scenarios. However, even as LLMs get increasingly more capable, it is unlikely that they will lead to translational solutions alone. Although many aspects are crucial for an NLP system's success, we use this work to highlight one key aspect of building real-world systems which is sometimes taken for granted: formulating a problem in a way that is well-aligned with its real-world counterpart. To explore the effect of this key step in building real-world NLP systems, we provide a case study on the important task of UMLS vocabulary insertion.

The Unified Medical Language System (UMLS) (Bodenreider, 2004) is a large-scale biomedical knowledge base that standardizes over 200 medical vocabularies. The UMLS contains approximately 16 million source-specific terms, referred

---

*Part of this work was done while interning at the NLM.

[1]Our code is available at https://github.com/OSU-NLP-Group/UMLS-Vocabulary-Insertion.

to as atoms, grouped into over 4 million unique concepts, making it one of the most comprehensive publicly available biomedical knowledge bases and a crucial resource for biomedical interoperability. Many of the vocabularies which make up the UMLS are independently updated to keep up with the rapidly advancing biomedical research field. In order for this essential public resource to remain up-to-date, a team of expert editors painstakingly identify which new atoms should be integrated into existing UMLS concepts or added as new concepts, as shown in Figure 1a. This process, which we refer to as UMLS vocabulary insertion (UVI), involves inserting an average of over 300,000 new atoms into the UMLS and is carried out twice a year before each new UMLS version release.

Despite its importance, scale and complexity, this task is accomplished by editors using lexical information (McCray et al., 1994), synonymy information provided by the source vocabularies and their own expertise. In order to improve this process, much work has been done to augment it with modern NLP techniques. In Nguyen et al. (2021), the authors introduce datasets and models which explore the task of UMLS vocabulary alignment (UVA). As seen in Figure 1b, the authors formulate the UVA task as a binary synonymy prediction task between two UMLS atoms, while the real-world task requires the whole UMLS to be considered and a concept to be predicted for each new atom (unless it is deemed a new concept atom). Unfortunately, while the UVA task has successfully explored biomedical synonymy prediction, its formulation has made it unable to yield practical improvements for the UVI process.

In this work, we attempt to address this gap with a novel UVI problem formulation, also depicted in Figure 1b. Our formulation follows the real-world task exactly by predicting whether a new atom should be associated with an existing concept or identified as a new concept atom. We introduce five datasets taken directly from actual UMLS updates starting from the second half of 2020 until the end of 2022. These datasets enabled us to measure the real-world practicality of our systems and led us to findings we could not have discovered otherwise. First, we find that adapting UVA models to perform the UVI task yields much higher error rates than in their original task, showing that their strong performance does not transfer to the real-world setting. Second, contrary to previous

work (Bajaj et al., 2022), we find that biomedical language models (LMs) outperform previous UVA models. Thirdly, we discover that rule-based and deep learning frameworks greatly improve each other's performance. Finally, inspired by biomedical entity linking and the complementary nature of our baseline systems, we propose a null-aware and rule-enhanced re-ranking model which outperforms all other methods and achieves low error rates on all five UMLS update datasets. To show our model's practical utility, we quantitatively evaluate its robustness across UMLS update versions and semantic domains, conduct a comparative evaluation against the second best method and carry out a qualitative error analysis to more deeply understand its limitations. We hope that our case study helps researchers and practitioners reflect on the importance of problem formulation for the translational success of NLP systems.

## 2 Related Work

### 2.1 UMLS Vocabulary Alignment

Previous work to improve UMLS editing formulates the problem as biomedical synonymy prediction through the UMLS vocabulary alignment task (Nguyen et al., 2021, 2022; Wijesiriwardene et al., 2022). These investigations find that deep learning methods are effective at predicting synonymy for biomedical terms, obtaining F1 scores above 90% (Nguyen et al., 2021). Although this formulation can help explore biomedical synonymy prediction, it does not consider the larger UMLS updating task and thus the strong performance of these models does not transfer to real-world tasks such as UVI.

Apart from the clear difference in scope between UVA and UVI shown in Figure 1b, major differences in evaluation datasets contribute to the gap in UVA's applicability to the UVI task. In Nguyen et al. (2021), the authors built a synonymy prediction dataset with almost 200 million training and test synonym pairs to approximate the large-scale nature of UMLS editing. UVA dataset statistics can be found in Appendix A. Since the UVA test set was created using lexical similarity aware negative sampling, it does not hold the same distribution as all the negative pairs in the UMLS. Since the UVI task considers all of the UMLS, UVA sampling leads to a significant distribution shift between these tasks. This unfortunately diminishes the usefulness of model evaluation on the UVA dataset for the real-world task. Surprisingly,

this gap results in biomedical language models like BioBERT (Lee et al., 2019) and SapBERT (Liu et al., 2021) underperforming previous UVA models in the UVA dataset Bajaj et al. (2022) while outperforming them in our experiments.

## 2.2 Biomedical Entity Linking

In the task of biomedical entity linking, terms mentioned within text must be linked to existing concepts in a knowledge base, often UMLS. Our own task, UMLS vocabulary insertion, follows a similar process except for three key differences: 1) relevant terms come from biomedical vocabularies rather than text, 2) some terms can be new to the UMLS and 3) each term comes with source-specific information. Many different strategies have been used for biomedical entity linking such as expert-written rules (D'Souza and Ng, 2015), learning-to-rank methods (Leaman et al., 2013), models that combine NER and entity-linking signals (Leaman and Lu, 2016; Furrer et al., 2022) and language model fine-tuning (Liu et al., 2021; Zhang et al., 2022; Yuan et al., 2022). Due to the strong parallels between biomedical entity-linking and our task, we leverage the best performing LM based methods for the UVI task Liu et al. (2021); Zhang et al. (2022); Yuan et al. (2022). These methods fine-tune an LM to represent synonymy using embedding distance, enabling a nearest neighbor search to produce likely candidates for entity linking.

The first difference between biomedical entity linking and UVI is addressed by ignoring textual context as done in Liu et al. (2021), which we adopt as a strong baseline. The second difference, that some new atoms can be new to the UMLS, is addressed by work which includes un-linkable entities in the scope of their task (Ruas and Couto, 2022; Dong et al., 2023). In these, a cross-encoder candidate module introduced by Wu et al. (2020) is used to re-rank the nearest neighbors suggested by embedding methods like Liu et al. (2021) with an extra candidate which represents that the entity is unlinkable, or in our case, a new concept atom. The third difference has no parallel in biomedical entity linking since mentions do not originate from specific sources and is therefore one of our contributions in §4.6.

## 3 UMLS Vocabulary Insertion

We refer to UMLS Vocabulary Insertion (UVI) as the process of inserting atoms from updated or new medical vocabularies into the UMLS. In this task, each new term encountered in a medical source vocabulary is introduced into the UMLS as either a synonym of an existing UMLS concept or as an entirely new concept. In this section, we describe our formulation of the UVI task, the baselines we adapted from previous work, as well as a thorough description of our proposed approach.

## 3.1 Problem Formulation

First, we define the version of the UMLS before the update as $K := \{c_1, ..., c_n\}$, a set of unique UMLS concepts $c_i$. Each concept $c_i$ is defined as $c_i := \{a_1^i, ..., a_{k_i}^i\}$ where each atom $a_j^i$, as they are referred to by the UMLS, is defined as the $j^{th}$ source-specific synonym for the $i^{th}$ concept in the UMLS.

In the UMLS Vocabulary Insertion (UVI) task, a set of $m$ new atoms $Q := \{q_1, ..., q_m\}$ must be integrated into the current set of concepts $K$. Thus, we can now define the UVI task as the following function $I$ which maps a new atom $q_j$ to its gold labelled concept $c_{q_j}$ if it exists in the old UMLS $K$ or to a null value if it is a new concept atom, as described by the following Equation 1.

$$I(K, q_j) = \begin{cases} c_{q_j} & \text{if } c_{q_j} \in K \\ \varnothing & \text{otherwise} \end{cases} \quad (1)$$

## 4 Experimental Setup

### 4.1 Datasets

To evaluate the UVI task in the most realistic way possible, we introduce a set of five insertion sets $Q$ which contain all atoms which are inserted into the UMLS from medical source vocabularies by expert editors twice a year. Due to their real-world nature, these datasets vary in size and new concept distribution depending on the number and type of atoms that are added to source vocabularies before every update as shown in Table 1. We note that the version of the UMLS we use contains $8.5$ rather than $16$ million atoms because we follow previous work and only use atoms that are in English, come from active vocabularies and are non-suppressible, features defined by UMLS editors.

While most of our experiments focus on the UMLS 2020AB, we use the other four as test sets to evaluate temporal generalizability. We split the 2020AB insertion dataset into training, dev and test sets using a 50:25:25 ratio and the other insertion datasets using a 50:50 split into dev and

| | Original UMLS $K$ | Insertion Set $Q$ | New Concepts |
|---|---|---|---|
| **2020AB** | 8,521,220 | 430,135 | 260,058 |
| **2021AA** | 8,839,907 | 226,210 | 91,834 |
| **2021AB** | 8,835,147 | 455,493 | 218,933 |
| **2022AA** | 9,175,923 | 175,989 | 111,853 |
| **2022AB** | 9,082,515 | 275,842 | 188,984 |

Table 1: UMLS Statistics from 2020AB to 2022AB. Our models are trained on the 2020AB insertion dataset.

test sets. We do stratified sampling to keep the distribution of semantic groups, categories defined by the UMLS, constant across splits within each insertion set. This is important since the distribution of semantic groups changes significantly across insertion datasets and preliminary studies showed that performance can vary substantially across categories. For details regarding the number of examples in each split and the distribution of semantic groups across different insertion sets, refer to Appendix B.

## 4.2 Metrics

We report several metrics to evaluate our methods comprehensively on the UVI task: accuracy, new concept metrics and existing concept accuracy.

**Accuracy.** It measures the percentage of correct predictions over the full insertion set $Q$.

**New Concept Metrics.** These measure how well models predict new atoms as new concepts and they are described in Equation 2. The terms in Equation 2, subscripted by *nc*, refer to the number of true positive (TP), false positive (FP) and false negative (FN) examples, calculated by using the new concept label as the positive class.

$$P_{nc} = \frac{TP_{nc}}{TP_{nc} + FP_{nc}}$$
$$R_{nc} = \frac{TP_{nc}}{TP_{nc} + FN_{nc}} \quad (2)$$

**Existing Concept Accuracy.** This metric shows model performance on atoms in $Q$ which were linked by annotators to the previous version of UMLS $K$, as shown in Equation 3. Let $N_{ec}$ be the number of concepts in $Q$ which were linked to concepts in $K$.

$$A_{ec} = \frac{1}{N_{ec}} \sum_{q_j \in Q} \begin{cases} \hat{c}_{q_j} = c_{q_j} & \text{if } c_{q_j} \in K \\ 0 & \text{otherwise} \end{cases} \quad (3)$$
$$\hat{c}_{q_j} := I(K, q_j)$$

## 4.3 UVA Baselines

We adapted several UVA specific system as baselines for our UMLS vocabulary insertion task.

**Rule-based Approximation (RBA). (Nguyen et al., 2021)** This system was designed to approximate the decisions made by UMLS editors regarding atom synonymy using three simple rules. Two atoms were deemed synonymous if 1) they were labelled as synonyms in their source vocabularies, 2) their strings have identical normalized forms and compatible semantics (McCray et al., 1994) and 3) the transitive closure of the other two strategies. We thus define the $I$ function for the UVI task as follows. We first obtain an unsorted list of atoms $a_i$ in $K$ deemed synonymous with $q_j$ by the RBA. We then group these atoms by concept to make a smaller set of unique concepts $c_i$. Since this predicted concept list is unsorted, if it contains more than one potential concept, we randomly select one of them as the predicted concept $\hat{c}_{q_j}$. If the RBA synonym list is empty, we deem the new atom as not existing in the current UMLS version.

**LexLM. (Nguyen et al., 2021)** The Lexical-Learning Model (LexLM) system was designed as the deep learning alternative to the RBA and trained for binary synonymy prediction using the UVA training dataset. Their proposed model consists of an LSTM encoder over BioWordVec (Zhang et al., 2019) embeddings which encodes two strings and calculates a similarity score between them. A threshold is used over the similarity score to determine the final synonymy prediction.

To adapt this baseline to the UVI task, we define the insertion function $I$ as mapping a new atom $q_j$ to the concept in $K$, $\hat{c}_{q_j}$, containing the atom with the highest similarity score to $q_j$ based on the LexLM representations. To allow the function $I$ to predict that $q_j$ does not exist in the current UMLS and should be mapped to the empty set $\varnothing$), we select a similarity threshold for the most similar concept under which $q_j$ is deemed a new atom. For fairness in evaluation, the similarity threshold is selected using the 2020AB UVI training set.

## 4.4 LM Baselines

Previous work finds that language models do not improve UVA performance (Bajaj et al., 2022). However, given our new formulation, we evaluate

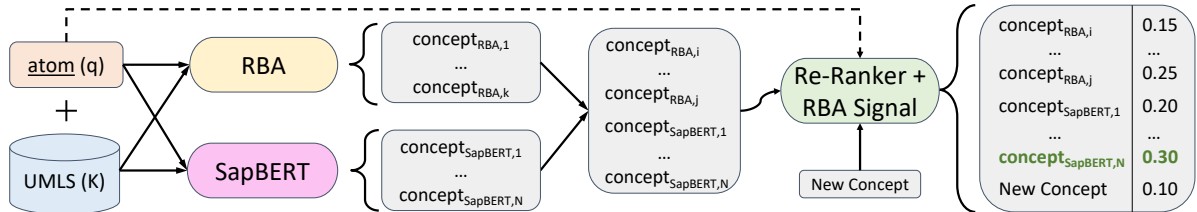

Figure 2: Overall architecture for our best performing approach on the new UVI task formulation. Our methodology leverages the best distance-based ranking model (SapBERT) as well as RBA signal. Additionally, our design allows new atoms to be identified as new concepts by introducing a 'New Concept' placeholder into the candidate list given to the re-ranking module as shown above.

two language models in the more realistic UVI task using the same strategy described for the LexLM model above. For implementation details, we refer the interested reader to Appendix C.

**PubMedBERT (Gu et al., 2021).** PubMedBERT is one of the most capable biomedical specific language models available due to its from scratch pre-training on biomedical data as well as its specialized biomedical tokenizer.

**SapBERT (Liu et al., 2021).** SapBERT is a language model designed for biomedical entity linking or concept normalization. It was developed by fine-tuning the original PubMedBERT on the 2020AA version of UMLS using a contrastive learning objective. This objective incentivizes synonymous entity representations in UMLS to be more similar than non-synonymous ones.

### 4.5 Augmented RBA

Given that the neural representation baselines discussed above provide a ranking system missing from the RBA, we create a strong baseline by augmenting the RBA system with each neural ranking baseline. In these simple but effective baselines, the concepts predicted by the RBA are ranked based on their similarity to $q_j$ using each neural baseline system. New concept prediction uses the same method employed by the original RBA model.

### 4.6 Our Approach: Candidate Re-Ranking

Our candidate re-ranking approach is inspired by some entity linking systems which use two distinct steps: 1) candidate generation, which uses a bi-encoder like the baselines described above, and 2) candidate re-ranking, in which a more computationally expensive model is used to rank the $k$ most similar concepts obtained by the bi-encoder. Other work (Wu et al., 2020) encodes both new

atoms and candidates simultaneously using language models, allowing for the encoding of one to be conditioned on the other. Our cross-encoder is based on PubMedBERT [2] and we use the most similar 50 atoms which represent unique concepts as measured by the best baseline, the RBA system augmented with SapBERT ranking. More concretely, the atom which represents each candidate concept $a_{c_i}$ is appended to new atom $q_j$ and encoded as follows: $[CLS]\ q_j\ [SEP]\ a_{c_i}$. Since the number of RBA candidates differs for every new atom, if the RBA produces less that 50 candidates, the remaining candidates are selected from SapBERT's nearest neighbor candidates. We use the BLINK codebase (Wu et al., 2020) to train our re-ranking module. More information about our implementation can be found in Appendix C.

#### 4.6.1 Null Injection

In contrast with standard entity linking settings where every mention can be linked to a relevant entity, UVI requires some mentions or new atoms to be deemed absent from the relevant set of entities. To achieve this in our re-ranking framework, we closely follow unlinkable biomedical entity linking methods (Dong et al., 2023; Ruas and Couto, 2022) and introduce a new candidate, denoted by the NULL token, to represent the possibility that the atom is new to the UMLS.

#### 4.6.2 RBA Enhancement

Finally, given the high impact of the RBA system in preliminary experiments, we integrate rule-based information into the candidate re-ranking learning. The RBA provides information in primarily two ways: 1) the absence of RBA synonyms sends a strong signal for a new atom being a novel concept in the UMLS and 2) the candidate concepts

---

[2] Preliminary results showed that PubMedBERT outperforms SapBERT as a re-ranker.

|  | Accuracy | New Concept | | | Existing Concept Accuracy |
| --- | --- | --- | --- | --- | --- |
|  |  | Recall | Precision | F1 |  |
| **Rule Based Approximation (RBA)** | 70.1 | 99.0 | 90.5 | 94.6 | 26.3 |
| **LexLM** | 63.2 | 89.5 | 92.4 | 90.9 | 22.4 |
| **PubMedBERT** | 68.4 | 99.1 | 67.3 | 80.2 | 20.7 |
| **SapBERT** | 77.4 | 94.1 | 79.2 | 86.0 | 52.0 |
| **RBA + LexLM** | 80.4 | 99.0 | 90.5 | 94.6 | 51.6 |
| **RBA + PubMedBERT** | 83.7 | 99.0 | 90.5 | 94.6 | 60.0 |
| **RBA + SapBERT** | 90.7 | 99.0 | 90.5 | 94.6 | 76.1 |
| **Re-Ranker (PubMedBERT)** | 85.5 | 96.3 | 91.6 | 93.9 | 68.4 |
| **+ RBA Signal** | 93.2 | 98.2 | 96.1 | 97.1 | 85.5 |

Table 2: Comparison for rule-based, distance-based and combined baselines against our re-ranking approaches both with and without RBA-signal over all our metrics. All results reported above were calculated on the 2020AB UMLS insertion dataset. We find that all improvements of our best approach over the RBA+SapBERT baseline are very highly significant (p-value < 0.001) based on a paired t-test with bootstrap resampling.

which the RBA predicted, rather than the ones predicted based solely on lexical similarity, have a higher chance of being the most appropriate concept for the new atom. Thus, we integrate these two information elements into the cross-encoder by 1) when no RBA synonyms exist, we append the string "(No Preferred Candidate)" to the new atom $q_j$ and 2) every candidate that was predicted by the RBA is concatenated with the string "(Preferred)". This way, the cross-encoder obtains access to vital RBA information while still being able to learn the decision-making flexibility which UMLS editors introduce through their expert knowledge.

## 5 Results & Discussion

In this section, we first discuss performance of our baselines and proposed methods on the UMLS 2020AB test set. We then evaluate the generalizability of our methods across UMLS versions and biomedical subdomains. Finally, we provide a comparative evaluation and a qualitative error analysis to understand our model's potential benefits and limitations.

### 5.1 Main Results

**Baselines.** As seen in Table 2, previous baselines such as RBA, LexLM and biomedical language models like PubMedBERT and SapBERT stay under the 80% mark in overall accuracy, with specially low performance in the existing concept accuracy metric. Even SapBERT, which is fine-tuned for the biomedical entity linking task, is unable to obtain high existing concept and new concept prediction scores when using a simple optimal similarity threshold method.

Nevertheless, a simple baseline which combines the strengths of neural models and the rule-based system obtains surprisingly strong results. This is especially the case for augmenting the RBA with SapBERT which obtains a num90% overall accuracy and existing concept accuracy of 76%. We note that the new concept recall and precision of all RBA baselines is the same since the same rule-based mechanism is used.

**Our Approach.** For the PubMedBERT-based re-ranking module, we find that the NULL injection mechanism enables it to outperform the models that rely solely on lexical information (LexLM, PubMedBERT and SapBERT) by a wide margin. However, it underperforms the best augmented RBA baseline substantially, underscoring the importance of RBA signal for the UVI task. Finally, we note that RBA enhancement allows the re-ranking module to obtain a 93.2% accuracy due to boosts in existing concept accuracy and new concept precision of almost 10% and 4% respectively. These improvements comes from several important features of our best approach which we discuss in more detail in §5.3, namely the ability to flexibly determine when a new atom exists in the current UMLS even when it has no RBA synonyms and to apply rules used by UMLS editors seen in the model's training data. This substantial error reduction indicates our method's potential as a useful tool for supporting UMLS editors.

### 5.2 Model Generalization

In this section, we note the robust generalization of our re-ranking module across both UMLS versions

and semantic groups (semantic categories defined by the UMLS).

**Across Versions.** In Figure 3, we see that the best performing baseline RBA + SapBERT and our best method obtain strong performance across all five UMLS insertion datasets. Even though our proposed approach obtains the largest gains in the 2020AB set in which it was trained, it achieves stable existing concept accuracy and new concept F1 score improvements across all sets and shows no obvious deterioration over time, demonstrating its practicality for future UMLS updates. Unfortunately, we do observe a significant dip in new concept F1 for all models in the 2021AA dataset mainly due to the unusually poor performance of the RBA in one specific source, Current Procedural Terminology (CPT), for that version.

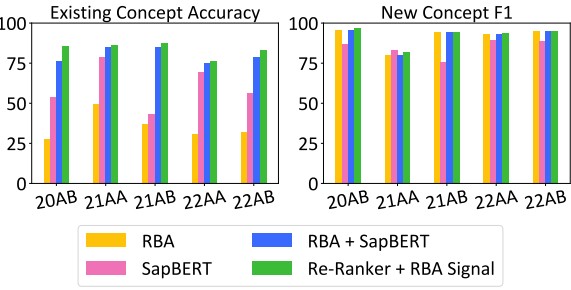

Figure 3: Existing concept accuracy (left) and new concept F1 (right) of the best model from each baseline type and our best approach across 5 UVI datasets from 2020AB to 2022AB. All improvements over the best baseline are very highly significant (p-value < 0.001).

**Across Subdomains.** Apart from evaluating whether our proposed approach generalizes across UMLS versions, we evaluate how model performance changes across different semantic groups. Table 3 shows the results of our best baseline (RBA + SapBERT) compared against our best proposed approach (Re-Ranker + RBA Signal) on the nine most frequent semantic groups averaged over all development insertion sets. We report the results in detail over all insertion sets in Appendix E. Our evaluation reveals that even though our best baseline performs quite well across several semantic groups, performance drops in challenging categories like *Drugs*, *Genes*, *Procedures* and the more general *Concepts & Ideas* category. Our approach is able to improve performance across most groups to above 90%, with the exception of *Genes* and *Procedures*. Since the distribution of semantic groups can vary widely across UMLS updates, as seen in

| Semantic Group | RBA + SapBERT | Re-Ranker + RBA Signal |
|---|---|---|
| **Living Beings** | 97.2 | 98.0 |
| **Chemicals & Drugs** | 81.1 | 93.7 |
| **Genes & Molecular Seq.** | 74.3 | 77.7 |
| **Disorders** | 92.1 | 97.7 |
| **Procedures** | 82.6 | 84.3 |
| **Physiology** | 92.8 | 99.0 |
| **Concepts & Ideas** | 89.1 | 97.2 |
| **Devices** | 90.7 | 97.4 |
| **Anatomy** | 95.1 | 98.3 |

Table 3: Accuracy by semantic group for the two highest performing UVI systems averaged over all development insertion sets from 2020AB to 2022AB.

the dataset details in Appendix B, our model's improved semantic group robustness is vital for its potential in improving the efficiency of the UMLS update process.

As for the categories in which our approach remained below 90% like *Genes* and *Procedures*, we find that they are mainly due to outlier insertion sets. Both the *Genes* and *Procedures* categories have one insertion set, 2022AA and 2021AA respectively, in which the performance of both systems drops dramatically due to a weak RBA signal which our methodology was unable to correct for. We refer the interested reader to Appendix E for these results and a more detailed discussion around this limitation.

### 5.3 Comparative Evaluation

As mentioned in the main results, our best model outperforms the best baseline mainly through improvements in existing concept accuracy and new concept precision. In Table 4, we report the distribution of 2,943 examples incorrectly predicted by RBA + SapBERT amended by our best approach. We note that a large majority, around 60%, of the corrections are *concept linking* corrections, new atoms which are linked to an existing concept correctly while they were wrongly predicted as new concept atoms by the baseline. Most of the remain-

| Correction Type | Correction % |
|---|---|
| **Concept Linking** | 59.5 |
| **Re-Ranking** | 35.9 |
| **New Concept Identification** | 4.6 |

Table 4: Distribution of examples incorrectly predicted by the best baseline amended by our best model.

| Correction Type | New Atoms | Top 5 RBA + SapBERT Candidates |
|---|---|---|
| Re-Ranking | Amorpha \<eudicots> | **Amorpha \<moth> (Preferred)** 
 *Amorpha \<angiosperm> (Preferred)* 
 Amorphus 
 Amorphus sp. 
 Amorphotheca |
| | Cytarabine-Thioguanine | **cytarabine/thioguanine (Preferred)** 
 *Cytarabine-Thioguanine Regimen (Preferred)* 
 cyclophosphamide/cytarabine/thioguanine 
 Cytarabine/Mitoxantrone/Thioguanine 
 Cytarabine/Doxorubicin/Thioguanine |
| Concept Linking | total hysterectomy with removal of right ovary | **[NEW CONCEPT]** 
 *Total hysterectomy with right oophorectomy* 
 abdominal hysterectomy with removal of right ovary 
 Total hysterectomy with right salpingo-oophorectomy 
 Total hysterectomy with removal of both tubes and ovaries |
| | WARFARIN NA (JANTOVEN) 7.5MG TAB | **[NEW CONCEPT]** 
 *Warfarin Sodium 7.5 MG Oral Tablet [JANTOVEN]* 
 WARFARIN NA (TARO) 7.5MG TAB 
 Warfarin Sodium 7.5 MG Oral Tablet [COUMADIN] |

Table 5: Some examples which were incorrectly predicted by our best baseline (RBA + SapBERT), shown above in **red**, but corrected by our best proposed re-ranking model, shown above in ***green***.

| Error Type | New Atom | Top 5 Best Model |
|---|---|---|
| UMLS Error (Duplicate Concepts) | Left orbital region | **Left orbital region (Preferred)** 
 *[NEW CONCEPT]* 
 Structure of periorbital region of left eye 
 Left orbital cavity proper 
 Left orbital content |
| | Gonostomatidae \<ciliates> | **Gonostomatidae (Preferred)** 
 *Gonostomatidae (Preferred)* 
 [NEW CONCEPT] 
 Gonichthys 
 Protrodiplostomatidae |
| True Errors | urea 400 MG/ML | **[NEW CONCEPT]** 
 Urea 400 mg/mL cutaneous lotion 
 urea@50 %@TOPICAL@SOLUTION 
 *urea 40%* 
 UREA 40% TOP GEL |
| | Exanthem caused by human echovirus (disorder) | **[NEW CONCEPT]** 
 VIRUSES ACCOMPANIED BY EXANTHEM 
 exanthems viral 
 Exanthem 
 *viral exanthem due to echovirus* |

Table 6: Some examples which were incorrectly predicted by our best proposed model, shown in **red**. Gold label concepts are marked with ***green***. The first two rows show two errors caused by UMLS annotations while the final two are legitimate errors caused by complexity and ambiguity.

ing corrections, 35.9%, are *re-ranking* corrections based on our model's ability to re-rank gold concept over other candidate concepts. The final 5% comes from *new concept identification* corrections in which a new atom is correctly identified as a new concept atom when it was incorrectly linked to an existing one by the best baseline.

The examples shown in Table 5 illustrate the benefits of our proposed approach more clearly. In the first two rows, we see two *re-ranking* corrections. In the first example, SapBERT incorrectly identifies '\<eudicots>' as being closer to '\<moth>' than '\<angiosperm>' but our model has learned to interpret the disambiguation tags and correctly associates 'eudicots' with 'angiosperm' as levels of plant family classifications. In the second example, we observe that our trained model learns to link new atoms to concepts which have more comprehensive information such as the addition of the "Regimen" phrase. Although this is an editorial rule rather than an objective one, it is important to note that our model can adequately encode these.

The final two rows in Table 5 show *concept linking* corrections. These examples illustrate the most important feature of our proposed model, the ability to link new atoms to concepts even when the RBA would consider them a new concept atom. In these instances, the model must determine whether all the features in the new atom are present in any potential candidates without support from the RBA. In these two examples, the model is able to correctly identify synonymy by mapping 'removal of the right ovary' to 'right oophorectomy', 'NA' to 'Sodium' and 'TAB' to 'Oral Tablet.

## 5.4 Error Analysis

Given that our work focuses on a specific practical application, in this section, we aim to more deeply understand how our approach can be effectively adopted by UMLS editors in their vocabulary insertion task. To this end, we recruited a biomedical terminology expert familiar with the UMLS vocabulary insertion process to analyze the practical effectiveness and limitations of our system.

We first studied the calibration of our best model's output as a way to understand its error detection abilities. As shown in detail in Appendix F, we see a substantial drop in performance when model confidence, a softmax over candidate logit scores, drops below 90%. This drop could indicate that our model is well calibrated, however, our qualitative experiments reveal that this signal comes from a large number of annotation errors in the UMLS which are easily detected by our problem formulation.

We discovered this through a qualitative error analysis carried out with the help of the aforementioned biomedical terminology expert. We chose three sets of 30 randomly chosen example errors with different model confidence scores: high (90%-100%), medium (60%-70%) and low (30%-40%). Our expert editor reports several important findings. First, there was no substantial difference in example difficulty between different model confidence bins. Second, 70% of model errors are caused by the existence of UMLS concepts which have

phrases that are equivalent to the new atoms, leading to ambiguous examples which can be found in the first section of Table 6. This arises from two types of annotation errors within the UMLS, either the new atom was incorrectly introduced into the UMLS or the phrase that is representing that concept was previouly introduced into UMLS incorrectly. Out of this study, the expert found 15 out of the 90 instances where our model's suggestions lead to detecting incorrect associations in the original UMLS vocabulary insertion process. This evaluation suggests that our model could be quite useful in supporting quality assurance for the UMLS.

Even though most model errors are caused by annotation issues in the UMLS, there are still some which are due to complexity and ambiguity. In the bottom half of Table 6, we see examples that our model still struggles with. First, the new atom "urea 400 MG/ML" should have been mapped to "urea 40%" since the percentage is calculated as the number of grams in 100 mL. However, this decision requires not only the knowledge of this definition but also mathematical reasoning abilities. Finally, the last error in our table is caused by the ambiguity in deciding whether "human echovirus" and "echovirus" should be deemed equivalent. We note that both of these error types as well as the previously mentioned annotation errors show that our model's errors are occurring on scenarios which are either unsolvable or very challenging, shedding light on its potential as a practical system to support UMLS editors.

## 6 Conclusion

In conclusion, this paper emphasizes the importance of formulating NLP problems that align well with real-world scenarios in the midst of growing enthusiasm for NLP technologies. Focusing on the real-world task of UMLS vocabulary insertion, we demonstrate the importance of problem formulation by showcasing the differences between the UMLS vocabulary alignment formulation and our own UVI formulation. We evaluate existing UVA models as baselines and find that their performance differs significantly in the real-world setting. Additionally, we show that our formulation allows us to not only discover straightforward but exceptionally strong new baselines but also develop a novel null-aware and rule-enhanced re-ranking model which outperforms all other methods. Finally, we show

that our proposed approach is highly translational by providing evidence for its robustness across UMLS versions and biomedical subdomains, exploring the reasons behind its superior performance over our baselines and carrying out a qualitative error analysis to understand its limitations. We hope our case study highlights the significance of problem formulation and offers valuable insights for researchers and practitioners for building effective and practical NLP systems.

## 7 Limitations

We acknowledge several limitations to our investigation, which we propose to address in future work. First, while our formulation aligns exactly with part of the insertion process, there are aspects of the full insertion of new terms into the UMLS which are out of our scope. While we do identify terms that are not linked to existing UMLS concepts, we do not attempt to group these terms into new concepts. The identification of synonymous terms for new concepts will be addressed in future work. Second, except for the RBA approach that leverages lexical information and source synonymy, our approach does not take advantage of contextual information available for new terms (e.g., hierarchical information provided by the source vocabulary). We plan to follow (Nguyen et al., 2022) and integrate this kind of information that has been shown to increase precision without detrimental effect on recall in the UVA task. Third, our approach uses a single term, the term closest to the new atom, as the representative for the concept for linking purposes. While this approach drastically simplifies processing, it also restricts access to the rich set of synonyms available for the concept. We plan to explore alternative trade offs in performance when including more concept synonyms. Finally, reliance on the RBA information had the potential for incorrectly identifying new concepts when RBA signal is not complete. Even though RBA signal is quite useful for this task, it is important to build systems robust to its absence. We plan to explore this robustness more actively in future work by including such incomplete signal in the training process.

## 8 Acknowledgements

The authors would like to thank the expert UMLS annotators from the NLM for their detailed error analysis. We also appreciate constructive comments from anonymous reviewers and our NLM

and OSU NLP group colleagues. This research was supported in part by NIH R01LM014199, the Ohio Supercomputer Center (Center, 1987) and the Intramural Research Program of the NIH, National Library of Medicine.

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

## A  Original UVA Dataset

Table 7 lists the basic statistics for the UMLS vocabulary alignment datasets. Since the UVA task was formulated and evaluated only as a binary classification task, the dataset is divided into positive and negative pairs. For more details about how the negative pairs were sampled from the UMLS, we refer the interested reader to §4.2 of Nguyen et al. (2021).

|  | UVA Pairs | Positive Pairs | Negative Pairs |
|---|---|---|---|
| **Train** | 192,400,462 | 22,324,834 | 170,075,628 |
| **Test** | 173,035,862 | 5,581,209 | 167,454,653 |

Table 7: Original UVA dataset statistics.

## B  UVI Dataset Details

In Table 8, we report the size of our five UMLS vocabulary insertion dataset splits. We note that only the 2020AB version contains a training set, all other insertion sets only have development and test sets.

|  | Train | Dev | Test |
|---|---|---|---|
| **2020AB** | 215,402 | 105,796 | 108,937 |
| **2021AA** | − | 112,647 | 113,563 |
| **2021AB** | − | 227,440 | 228,053 |
| **2022AA** | − | 88,186 | 87,803 |
| **2022AB** | − | 138,107 | 137,735 |

Table 8: Experimental split statistics for UMLS insertion dataset $Q$ from 2,020 to 2,022.

In terms of dataset construction, we reiterate that stratified sampling based on semantic groups was used to keep the original distributions intact. We adopt this technique due to the substantial changes

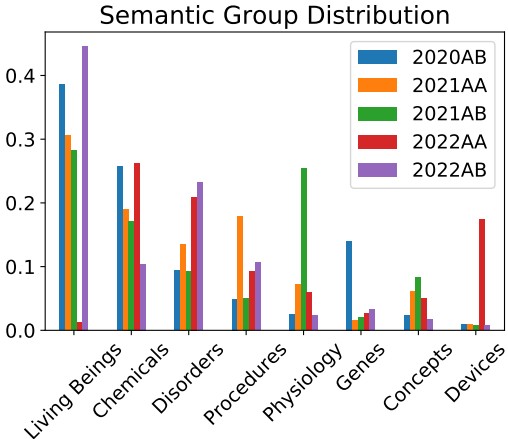

Figure 4: This figure shows the incidence of each of the most frequent 8 semantic groups across the 5 insertion sets explored in this work.

in semantic group distribution across insertion sets, as seen in 4, as well as the high variance in model performance across semantic categories, as seen in §5.2 and Appendix E.

## C  Implementation Details

In this section we discuss the implementation details for our baselines as well as our proposed approach. For the UMLS vocabulary alignment baselines, we use the same implementation of the Rule-Based Approximation (RBA) and LexLM used by the authors in Nguyen et al. (2021). To implement our language model baselines we use the HuggingFace Transformers library (Wolf et al., 2020). We use the FAISS library (Johnson et al., 2021) to speed up nearest neighbor search using GPUs when experimenting with LexLM, SapBERT and PubMedBERT embeddings (Johnson et al., 2021). We train our cross-encoder re-ranking module using BLINK (Wu et al., 2020), which uses a cross-entropy loss to maximize the score of the correct candidate over the rest of the candidates. We use default hyperparameters listed in Table 9 to train our re-ranking module but perform early stopping using the accuracy metric on our 2020AB validation set. All experiments used an NVIDIA V100 GPU with 16 GB of VRAM. The models we used and the approximate amount of GPU hours used for each is listed in Table 10.

| Learning Rate | Total Epochs | Batch Size | Warmup Ratio |
|---|---|---|---|
| 2e−5 | 3 | 1 | 0.1 |

Table 9: Hyperparameters selected for our cross-encoder re-ranking training for reproducibility.

|  | # of Parameters (millions) | Total GPU Hours |
|---|---|---|
| **LexLM** | 0.2 | 5 |
| **PubMedBERT** | 100 | 140 |
| **SapBERT** | 100 | 40 |

Table 10: Total GPU Hours associated with our experiments. PubMedBERT GPU hours include both UMLS encoding and fine-tuning for our re-ranking module.

## D  Latency Comparison

In Table 11, we report the inference latency for each baseline as well as our proposed approaches

on the UVI task. As seen in the table, our approach has significantly slower inference than previous baselines. Nevertheless, since the UMLS insertion task happens only twice a year, variations in inference latency are not a significant concern as long as the process can be run within a reasonable amount of time on available computing resources. We hope that these numbers can help other researchers and practitioners understand the computing requirements on this or similar tasks.

| Model | Inference Latency (ms) | Time for 300k Atoms (mins) |
|---|---|---|
| **RBA** | 0.01 | 0.05 |
| **LexLM** | 1.28 | 6.40 |
| **SapBERT** | 2.50 | 12.50 |
| **RBA + LexLM** | 1.29 | 6.45 |
| **RBA + SapBERT** | 2.51 | 12.55 |
| **Re-Ranker (RBA Signal)** | 35.51 | 177.5 |

Table 11: Time spent on inference for each baseline as well as our proposed approach.

## E  Detailed Semantic Group Evaluation

As mentioned in 5.2, different UMLS updates often contain completely different semantic group distributions since they depend entirely on independent source updates. Due to this, generalization across different semantic categories (semantic groups in the UMLS) is a crucial feature for a system to be successful in real-world UMLS vocabulary insertion. Table 13 provides a detailed report of the performance of our strongest baseline and our best proposed approach on all development insertion sets across the 9 most frequent semantic groups. As seen in these detailed results, our proposed approach obtains stronger and more consistent results across all semantic groups compared to our best baseline.

Nevertheless, as discussed in the main text, our approach remained below 90% on average in categories like *Genes* and *Procedures*. In the broken down results in Table 13, we can more clearly see that these averaged results are caused by outlier insertion sets. For the *Genes* semantic group, our proposed approach improves performance considerably for all insertion sets except for 2022AA, in which its performance drops by more than 10 points. We note that the performance of the best baseline is also much lower than usual, potentially indicating a weak RBA signal and challenging

atoms to link. For the *Procedures* category, we see a similar pattern in the 2021AA insertion set while the other sets see small but regular improvements with our system. These results indicate that, although our proposed approach can leverage the RBA signal more consistently when it is sufficiently strong, it fails to correct for it when it is very weak to begin with. It is therefore important to continue working on ways to correct or at least alert annotators about potential system failures in specific concept sub-groups.

## F  Model Calibration Details

As discussed above, our re-ranker model's output confidence, defined as a softmax over candidate logit scores produced by our model, seemed correlated with model accuracy. In Table 12, we show model accuracy across different model confidence scores. We find that model confidence score is highly correlated with model accuracy, which drops to around 50% when model confidence drops below 90% and continues to drop after that. Through qualitative analysis, we find that this does not indicate successful model calibration but is actually mainly caused by annotation errors within UMLS which result in duplicate and ambiguous concepts.

| Model Confidence (%) | Number of Examples | Accuracy |
|---|---|---|
| **0** | 23 | 8.7 |
| **10** | 80 | 22.5 |
| **20** | 206 | 32.5 |
| **30** | 397 | 36.0 |
| **40** | 1,282 | 56.0 |
| **50** | 964 | 55.1 |
| **60** | 511 | 48.7 |
| **70** | 411 | 50.6 |
| **80** | 590 | 55.3 |
| **90** | 38,076 | 92.1 |
| **100** | 62,743 | 99.8 |

Table 12: The output probability of our best re-ranking approach (the probability of the highest scoring candidate concept) seemed to be correlated with high prediction accuracy but actually indicates annotation errors.

| Semantic Group | 2020AB | | 2021AA | | 2021AB | | 2022AA | | 2022AB | |
|---|---|---|---|---|---|---|---|---|---|---|
| | RBA + SapBERT | Re-Ranker + RBA Signal | RBA + SapBERT | Re-Ranker + RBA Signal | RBA + SapBERT | Re-Ranker + RBA Signal | RBA + SapBERT | Re-Ranker + RBA Signal | RBA + SapBERT | Re-Ranker + RBA Signal |
| **Living Beings** | 99.2 | 99.8 | 98.6 | 95.8 | 96.8 | 99.7 | 93.3 | 95.3 | 97.9 | 99.6 |
| **Chemicals & Drugs** | 87.7 | 94.8 | 73.8 | 89.2 | 87.9 | 95.3 | 81.5 | 96.6 | 74.8 | 92.4 |
| **Genes & Molecular Sequences** | 86.8 | 97.0 | 76.2 | 82.6 | 78.2 | 87.4 | 58.9 | 42.5 | 71.2 | 79.2 |
| **Disorders** | 91.7 | 98.0 | 90.2 | 97.2 | 96.0 | 98.3 | 91.8 | 97.0 | 90.8 | 98.0 |
| **Procedures** | 94.1 | 96.9 | 54.6 | 54.8 | 95.3 | 97.6 | 95.0 | 97.0 | 74.2 | 75.0 |
| **Physiology** | 95.1 | 99.2 | 98.8 | 98.9 | 84.3 | 99.1 | 97.1 | 99.3 | 88.7 | 98.4 |
| **Concepts & Ideas** | 91.6 | 97.4 | 70.5 | 96.1 | 98.4 | 98.5 | 92.6 | 96.4 | 92.5 | 97.5 |
| **Devices** | 93.4 | 97.8 | 89.4 | 95.5 | 94.3 | 97.1 | 90.3 | 99.7 | 86.2 | 96.9 |
| **Anatomy** | 92.7 | 96.4 | 94.2 | 97.9 | 92.2 | 98.4 | 98.3 | 99.0 | 97.8 | 99.4 |

Table 13: Breakdown for Table 3 over all insertion development sets and the 9 most frequent semantic groups. These detailed results can help us more closely understand model failures across semantic groups compared to the aggregated results.