# OpenReview forum: "Solving the Right Problem is Key for Translational NLP: A Case Study in UMLS Vocabulary Insertion"
_EMNLP/2023/Conference — EMNLP 2023 Findings_

### Official Review · Reviewer_F1a9 · 2023-08-01

**Soundness:** 4

**Excitement:**

3: Ambivalent: It has merits (e.g., it reports state-of-the-art results, the idea is nice), but there are key weaknesses (e.g., it describes incremental work), and it can significantly benefit from another round of revision. However, I won't object to accepting it if my co-reviewers champion it.

**Paper Topic And Main Contributions:**

The focus of this work is better integration of realistic scenarios in the task of UMLS vocabulary insertion. The authors introduce the UVI task of how a new UMLS atom can be inserted - if it is related to an existing concept or if it is a new concept altogether. The authors contribute five datasets of UMLS updates over a period of time for the task. They integrate domain knowledge by using biomedical LMs. They also propose a model for the task and show through comparisons with baselines and across the datasets the improvement brought by the model.

This is actually a nicely written paper and reads well. I think with some revisions this paper should be accepted

**Questions For The Authors:**

1. Lines 067-068: How do they know about this number 300,000 related to UVI (unless of course, they work at NIH)? Providing a citation will significantly strengthen the claim and the motivation for the work. Also, Table 1 shows the statistics of the 5 datasets used with the Insertion sets which likely average to 300,000, however, can this be generalized beyond these 5?

2. Lines 243-244: Was there any rationale for choosing the 2020AB dataset for training?

3. Lines 248-252: The authors mention significant performance variability across concept categories. How different are these semantic categories across the five datasets?

4. Lines 390-393: How is the higher chance of being the most appropriate concept for the new atom determined? Any citation for this?

5. Lines 475-492: Can these findings be generalized for the other datasets?

**Reasons To Accept:**

1. The authors motivated the work nicely. They discuss the research gap in detail and how their work attempts to address this gap. The section on problem formulation defines the problem of UVI clearly.

2. Thorough experiments - comparison with baselines and across datasets for generalization.

3. Qualitative error analysis was done by experts

**Reasons To Reject:**

1. The proposed approach is shown to have higher accuracy/f1, but no statistical significance is provided.

2. Findings from comparison across subdomains (Lines 475-492) are provided in the appendix. This looks like an important result since the authors mentioned significant performance variations across semantic categories. Also in the appendix data is provided only for the 2020AB dataset.

**Reproducibility:**

3: Could reproduce the results with some difficulty. The settings of parameters are underspecified or subjectively determined; the training/evaluation data are not widely available.

**Reviewer Confidence:**

3: Pretty sure, but there's a chance I missed something. Although I have a good feel for this area in general, I did not carefully check the paper's details, e.g., the math, experimental design, or novelty.

---

> ### Author Rebuttal · Authors · 2023-08-28
>
> We are pleased that the reviewer considers our work to be nicely written, clearly motivated, experimentally thorough and that the high-quality of our error analysis was appreciated. We are very grateful for the time and effort the reviewer spent evaluating our paper as well as their thoughtful questions and concerns, as they will surely improve the quality of our work.
>
> We hope to address their concerns in the revisions discussed below:
> \
> \
> **Significance Testing**
>
> We thank the reviewer for pointing out the lack of significance testing. We carry out a Paired T-Test using Bootstrap Resampling and find that all improvements for our approach over the best baseline (RBA+SapBERT) are very highly significant (p-value < 0.001) on all metrics. This includes improvements in our generalization experiments in Figure 3. We will add this to the paper and note it in Table 2 and Figure 3.
> \
> \
> **Subdomain Variance**
>
> *“Lines 248-252: The authors mention significant performance variability across concept categories. How different are these semantic categories across the five datasets?”*
>
> We will add a plot with the semantic category distribution across versions to our paper to illustrate their high variability. To give the reviewer a better idea of this variance across insertion sets in this rebuttal, we note that the coefficient of variation for the most frequent semantic groups ranges between 33% and 110%. These large values indicate that the standard deviation compared to the mean percentage of each semantic group is high.
>
> *“Findings from comparison across subdomains (Lines 475-492) are provided in the appendix. This looks like an important result since the authors mentioned significant performance variations across semantic categories. Also in the appendix data is provided only for the 2020AB dataset.”*
>
> We concur with the reviewer that Table 10, which shows the subdomain performance variation in 2020AB, would be a good addition to the main paper. We will include the aggregate of subdomain performance variation across all UMLS updates and include a larger table with the data for each insertion set separately in the appendix for clarity.
>
> *“Lines 475-492: Can these findings be generalized for the other datasets?”*
>
> Our analysis concerning the low performance of the baseline in the “Genes” and “Drugs” categories as well as the strong improvement of our approach generalizes across all insertion sets except for “Genes” in 2022AA. In this update, the best baseline’s performance on “Genes” is much worse than in other updates due to a weak RBA signal. This domain shift could explain the lack of improvements from our approach in this specific instance. We will discuss this analysis and explanation in the camera-ready version of our paper.
> \
> \
> **Main Update Choice**
>
> *“Lines 243-244: Was there any rationale for choosing the 2020AB dataset for training?”*
>
> We chose the 2020AB dataset as our starting point for this study since it is the earliest possible version for which we are able to fairly compare with related baselines. Given that both the LexLM model and SapBERT were trained on large sections of the 2020AA version of UMLS, we could not choose any update prior to 2020AB. On the other hand, choosing any update after 2020AB for training would reduce the number of datasets we could use to test for model generalization across versions.
> \
> \
> **Insertion Set Size Information**
>
> We calculated the average number of new atoms as 300,000 as the mean of all 5 insertion sets we study in the paper. A more long-term average for new english atom which ignores filters described in Section 4.1 can be computed from the “Statistics” section of each release taken from the publicly available “Release Documentation Archive” of the UMLS (https://www.nlm.nih.gov/research/umls/archive/archive_home.html). Using this resource older and post-2018 updates the average is still high but decreases to around 200,000 atoms. We will include this description in the camera-ready version.
> \
> \
> **RBA Signal Clarification**
>
> *“Lines 390-393: How is the higher chance of being the most appropriate concept for the new atom determined? Any citation for this?”*
>
> In lines 390-393, we mention how the RBA can go beyond lexical similarity in indicating whether an atom should be linked to a specific concept. This difference comes from the RBA’s use of source synonymy as described in section 4.3. This signal lets the RBA check if the new atom has a source-determined synonym already in the UMLS which makes predictions more accurate in cases where these synonyms are available. We understand the reviewer’s confusion and will clarify this accordingly.

---

### Official Review · Reviewer_SKdh · 2023-08-04

**Soundness:** 3

**Excitement:**

3: Ambivalent: It has merits (e.g., it reports state-of-the-art results, the idea is nice), but there are key weaknesses (e.g., it describes incremental work), and it can significantly benefit from another round of revision. However, I won't object to accepting it if my co-reviewers champion it.

**Missing References:**


Relevant to biomedical entity linking, section 2.2

Furrer L, Cornelius J, Rinaldi F. Parallel sequence tagging for concept recognition. BMC Bioinformatics. 2022 Mar 24;22(Suppl 1):623. doi: 10.1186/s12859-021-04511-y. PMID: 35331131; PMCID: PMC8943923.

**Paper Topic And Main Contributions:**


This paper deals with a practical problem, namely the regular updates of the UMLS, which involve the integration of new terminological vocabularies. The papers proposes a methodology which could be used to speed up the updates, by automatically finding whether
a new term is a potential synonymous of a term already in the UMLS, or it can be attributed to a novel concept, not yet present in the UMLS.

The paper proposes a different conceptualization of the problem compared to previous approaches. Rather then simply evaluating the similarity of a new term to existing terms, the paper proposes an approach where each new term is assessed in relation to the entire
UMLS, and either a concept is found to which the term can be assigned, or the term is maked as novel, and a new concept will have
to be created. I am not completely convinced that this different conceptualization can be considered innovative, as it seems to me to
be entirely derivable from the original one.




**Questions For The Authors:**


Please provide a more accurate description of the evaluation metrics in section 4.2.  I suggest to use formulas.

In particular the "ranking accuracy", which is a central metric in the paper because it is the one where the most
improvements are seen, is very superficially defined. It is not clear if it is a proper ranking metric, as the name
suggest, or a measure of accuracy. If it is a ranking metric, please explain in which sense it provides a measure
of the quality of the ranking. If it is not, please name it differently.

I also struggled a bit to understand the definition of New Concept Precision" because it is not immediately
obvious what is the different between "correct new concept predictions" and "true new concept atoms", but
I believe I understood it.

In any case, formulas would help understanding and remove some ambiguity.

Figure 2, not clear what the "New Concept" as input to the Re-Ranker represents.






**Reasons To Accept:**


Well developed study of the problem of UMLS update, which is framed as a problem similar but not identical to (biomedical) entity linking.
Well designed experimental setup, with one particular UMLS update used as a reference for training/development/testing, and other
updates used for further testing of the results. Interesting combination of rule-based and BERT-based methods. Good evaluation.


**Reasons To Reject:**


The problem tackled by the authors is extremely specific, and has a very narrow application, although the methods could probably be generalized to similar problems of knowledge base update.

The evaluation metrics are not sufficiently clearly described, but it might be just a question of providing a more formal definition.

**Reproducibility:**

4: Could mostly reproduce the results, but there may be some variation because of sample variance or minor variations in their interpretation of the protocol or method.

**Reviewer Confidence:**

5: Positive that my evaluation is correct. I read the paper very carefully and I am very familiar with related work.

**Typos Grammar Style And Presentation Improvements:**


line 127-129 emphasis inappropriate in my view

---

> ### Author Rebuttal · Authors · 2023-08-28
>
> We would first like to thank the reviewer for the time and effort spent assessing our paper. We are glad that the reviewer deems our work well-developed, our experiments well-designed and that they find our methodology for combining rule-based and neural models interesting.
>
> We will address the reviewer’s concerns regarding our work’s contribution, scope and evaluation metric clarity in the following sections:
>
> **Contribution & Scope**
>
> Our work strives to provide a real-world case study which highlights the importance of problem formulation in the successful development of NLP applications. In the case of UVI,  slightly different problem formulations in previous works led to the development of datasets, models and conclusions which were not well-aligned with the real-world task and were therefore less practically applicable. The effects of this gap serves to remind readers that problem framing is a vitally important part of NLP practice and deserves more attention in the applied NLP literature.
>
> In addition to this more abstract contribution,  as the reviewer correctly points out, our methodological contributions are directly applicable to others working in similar problems such as other knowledge base update tasks. Furthermore, as the reviewer also mentions in their “Reasons to Accept”, our work provides an interesting approach for integrating rule-based and neural signals for this task successfully. Our approach provides evidence that simple methods can be used to augment neural NLP models with external rule-based signals. Given the amount of signal which does not come from the text itself in real-world NLP applications, this insight can be quite valuable to the applied NLP community.
>
> **Metric Clarity**
>
> We greatly appreciate the reviewer’s constructive criticism on our metric descriptions, clarifying this important point is crucial for our work. We will be sure to add formulas and more thorough descriptions for each metric in the camera-ready version.
>
> In particular, ranking accuracy is the percentage of correct predictions computed only on atoms in the insertion set which exist within the previous UMLS version (equivalent to “Accuracy” except that “new concept atoms” are excluded). Since it is a standard “Accuracy” metric rather than a ranking based metric, we will take the reviewer’s suggestion and change its name to “Existing Concept Accuracy” throughout the paper for clarity.
>
> **“Figure 2, not clear what the "New Concept" as input to the Re-Ranker represents.”**
>
> The “New Concept” block in Figure 2 represents that our approach allows the re-ranking module  to label an atom as a “New Concept” instead of as one of the candidate concepts. We will make this much clearer in the caption and the figure itself in the camera-ready version.
>
> **“line 127-129 emphasis inappropriate in my view”**
>
> We concur with the reviewer that the emphasis is somewhat inappropriate and we will remove it.
>
> **Missing Reference**
>
> We thank the reviewer for bringing our attention to this important work, we will include it in our camera-ready version.

---

### Official Review · Reviewer_wXDg · 2023-08-05

**Soundness:** 4

**Excitement:**

4: Strong: This paper deepens the understanding of some phenomenon or lowers the barriers to an existing research direction.

**Missing References:**

Adding a citation for L. 90 "prediction, its formulation has made it unable to yield practical improvements for the UVI process." can be very nice.

**Paper Topic And Main Contributions:**

The authors present a new SOTA model for solving UMLS vocabulary insertion (UVI) and helping experts editors in their tasks. The paper start by introducing the UVI task and introduce its multiple contributions (language models, entity linking, candidate re-ranking, augmented RBA, NULL-injection and RBA enhancement). Then, they give context about previous contributions to the subject, why they are for most of them treating the problem in a very different manner than the one from real world usages and each pro & cons on them. Finally, they are concluding by the results, a deep error analysis and a study of model generalization over time and subjects.

**Questions For The Authors:**

Do you have any idea of the latency (inference time in ms) of your proposed systems ? And how does it compare to other systems ?

A question about L. 238, have you tried your systems on other languages to know if it is replicable on them ? Or if the specificities of other languages make it more complicated to transfer performances ? For example, Chinese, Hebrew, French or Turkish ?


**Reasons To Accept:**

The paper is well written and present extremely well the subject of study despite being a difficult task. They introduce the previous works perfectly and share constantly their motivations behind this work without denigrate any previous contributions.

The different experiments are interesting and give extraordinary results compared to the selected baselines. The best proposed system give constant and reliable performances across a large set of datasets curated from a large time frame (5 versions). The expert analysis give good hints about the reasons why the system are sometime badly performing and will be interesting improvements for the future.

Evaluating both accuracy and ranking accuracy is a very good way of combining both academia standard metrics and real-world applications metrics, since the tool can be used as an assistance tool for experts editors.


**Reasons To Reject:**

None

**Reproducibility:**

4: Could mostly reproduce the results, but there may be some variation because of sample variance or minor variations in their interpretation of the protocol or method.

**Reviewer Confidence:**

4: Quite sure. I tried to check the important points carefully. It's unlikely, though conceivable, that I missed something that should affect my ratings.

**Typos Grammar Style And Presentation Improvements:**

L. 608 "While models like ChatGPT have vast potential, they are unlikely to single-handedly provide translational NLP solutions.". I found this remark inappropriate according to the thematic of the paper. You are not evaluating LLMs and ChatGPT model. It looks more like a buzzword than to be an interesting remark.

---

> ### Author Rebuttal · Authors · 2023-08-28
>
> We are delighted that the reviewer found our paper well-presented, strongly motivated and interesting. Their recognition of our strong empirical results and thorough qualitative error analysis is especially encouraging. We would like to thank the reviewer for the amount of effort they spent familiarizing themselves with and evaluating our work. We address their questions in the sections below:
>
>  **Inference Latency**
>
> Given that the UVI process happens only twice a year, variations in inference latency are not a significant concern as long as the process can be run within a reasonable amount of time on standard computing resources.
>
> Nevertheless, we concur with the reviewer that this information could be empirically valuable for other tasks and datasets. We will thus add the table below listing the inference latencies of all baselines and our own approach.
>
> | Model                               | Inference Latency (ms) | Total Time for 300,000 Atoms (minutes) |
> | --------------------------------  | ------------------------------- | ---------------------------------------------------- |
> | RBA                                 | 0.01                               | 0.05                                                        |
> | LexLM                              | 1.28                               | 6.40                                                        |
> | SapBERT                         | 2.50                               | 12.50                                                      |
> | RBA + LexLM                  | 1.29                               | 6.45                                                         |
> | RBA + SapBERT             | 2.51                               |12.55                                                        |
> | Re-Ranker (RBA Signal) | 35.51                             |177.5                                                        |
>
> **Multilingual Generalization**
>
> Although we have not tried our approach for other languages within the UMLS Metathesaurus, it is likely to generalize successfully as long as a strong language model which captures that language is used. However, performance will be limited by the amount of training data available since only a small percentage of all terms in the UMLS Metathesaurus are in languages other than English (Spanish is the second most common language with 1.9M terms and all other languages consist of fewer than 50,000 terms each).

---

### Meta-Review · Area_Chair_tK3g · 2023-09-06

**Recommendation:** 3

**Metareview:**

This paper is well-written, presents the subject clearly, provides strong experimental results, combines academic and real-world metrics, and offers valuable insights, though some aspects of the evaluation and findings could be improved.

---

### Decision · Program_Chairs · 2023-10-07

**Decision:**

Accept-Findings

**Comment:**

This paper is well-written, presents the subject clearly, provides strong experimental results, combines academic and real-world metrics, and offers valuable insights, though some aspects of the evaluation and findings could be improved.